# The Neural Bases of Egocentric Spatial Representation for Extracorporeal and Corporeal Tasks: An fMRI Study

**DOI:** 10.3390/brainsci11080963

**Published:** 2021-07-22

**Authors:** Stephanie Leplaideur, Annelise Moulinet-Raillon, Quentin Duché, Lucie Chochina, Karim Jamal, Jean-Christophe Ferré, Elise Bannier, Isabelle Bonan

**Affiliations:** 1Physical and Rehabilitation Medicine Department, Rennes University Hospital, 35000 Rennes, France; krmjamal@me.com (K.J.); isabelle.bonan@chu-rennes.fr (I.B.); 2Univ Rennes, CNRS, Inria, Inserm, IRISA UMR 6074, Empenn ERL U 1228, 35000 Rennes, France; quentin.duche@irisa.fr (Q.D.); jean-christophe.ferre@chu-rennes.fr (J.-C.F.); elise.bannier@irisa.fr (E.B.); 3Physical Medicine and Neurological Rehabilitation Department, Kerpape, 56270 Ploemeur, France; lucie.chochina@mutualite29-56.fr; 4Department of Physical and Rehabilitation Medicine, Duchenne Hospital, 62200 Boulogne sur Mer, France; anneliseraillon@yahoo.fr; 5Radiology Department, Rennes University Hospital, 35000 Rennes, France

**Keywords:** spatial cognition, egocentric reference frame, functional MRI

## Abstract

(1) Background: Humans use reference frames to elaborate the spatial representations needed for all space-oriented behaviors such as postural control, walking, or grasping. We investigated the neural bases of two egocentric tasks: the extracorporeal subjective straight-ahead task (SSA) and the corporeal subjective longitudinal body plane task (SLB) in healthy participants using functional magnetic resonance imaging (fMRI). This work was an ancillary part of a study involving stroke patients. (2) Methods: Seventeen healthy participants underwent a 3T fMRI examination. During the SSA, participants had to divide the extracorporeal space into two equal parts. During the SLB, they had to divide their body along the midsagittal plane. (3) Results: Both tasks elicited a parieto-occipital network encompassing the superior and inferior parietal lobules and lateral occipital cortex, with a right hemispheric dominance. Additionally, the SLB > SSA contrast revealed activations of the left angular and premotor cortices. These areas, involved in attention and motor imagery suggest a greater complexity of corporeal processes engaging body representation. (4) Conclusions: This was the first fMRI study to explore the SLB-related activity and its complementarity with the SSA. Our results pave the way for the exploration of spatial cognitive impairment in patients.

## 1. Introduction

Multiple sources of afferent information (from vestibular, visual, and proprioceptive receptors) and efferent information (from motor effectors) are needed to elaborate internal spatial representations of the body in space [1,2]. These inputs can be integrated into different and changing systems of coordinates, depending on the person’s position in space and the nature of the task. The egocentric reference frame encodes spatial information centered on body coordinates [3,4,5,6,7,8]. This frame involves two spatial components, depending on the intended action: extracorporeal and corporeal [9]. The extracorporeal component enables individuals to locate stimuli perceived in the extrapersonal space relative to their body and orient their actions in the environment. The corporeal component allows individuals to elaborate personal representations of the location and configuration of their body centered in space, in order to perform tasks such as personal care [4,10]. Two tasks have been used to explore these egocentric reference frames: the subjective straight-ahead task (SSA) and, more recently, the subjective longitudinal body plane task (SLB). The SSA explores extracorporeal space perception centered on the body. Participants imagine a plane starting from their midsagittal body and going straight ahead, dividing the extracorporeal space into two parts [4,10,11]. In the SLB, participants are asked to divide their corporeal space into two parts. They indicate with a button press when the bar virtually divided their corporeal space into two equal parts along their body’s midsagittal plane [5,11,12,13,14]. There are well-known clinical differences between extracorporeal and corporeal tasks [5,12], and the integration of relevant information can be differentially affected following brain damage [15]. For example, differences between extracorporeal and corporeal abilities have been identified when assessing unilateral spatial neglect. An impact of body representation in space on asymmetrical postural behaviors has also been reported. One previous study highlighted a strong relationship between SLB disturbance and postural asymmetry in patients with chronic stroke [5]. However, the neural basis of these differences has never been investigated. It is essential to improve knowledge in this field, as gaining a better understanding of the mechanisms involved in elaborating the egocentric reference frame would inform rehabilitation strategies for patients with spatial cognition disorders following stroke.

The present work was the first to use fMRI to study the commonalities and differences in brain activity between two egocentric tasks involving either extracorporeal or corporeal space. We hypothesized that both tasks would activate a bilateral fronto-parieto-occipital network, especially the posterior parietal and premotor cortices, with right hemispheric predominance [15,16,17,18]. Moreover, we expected the two tasks to activate distinct subregions of the network. This research among healthy participants was an ancillary study to a multicenter study exploring spatial cognitive disorders in patients with stroke.

## 2. Materials and Methods

### 2.1. Participants

We included 17 healthy volunteers (7 men and 10 women: Mean age = 50.4 years, SD = 14.9, range 23–70). They were all right handed on their declaration. All participants gave their written informed consent to take part in this study, which was approved by the Poitiers-France III West institutional review board (no. 05/12/16; AVCPOSTIM study—clinical trial identifier NCT01677091). Exclusion criteria were (1) contraindication to magnetic field exposure, (2) neurological disease, and (3) history of balance disorders. All participants had normal or corrected-to-normal vision. 

### 2.2. Experimental Design

The SSA and SLB tasks were performed in a supine position in an MRI scanner. For the SSA task, participants were instructed to divide the extracorporeal space in front of them into two halves [5,10,12,19]. A vertical green bar scanned the black screen from left to right and from right to left, with different random starting positions within the range of −7° to +7°, at a speed of 2° per second [20]. The instruction was “Press the button when the bar is straight ahead and divides the space in front of you into two equal parts” (Figure 1 and Appendix A).

During the SLB task, a green bar moved along the plane orthogonal to the line going through the navel [5,13,21]. The bar rotated alternately leftward and rightward, with different random starting angles within the range of −30°/+30°, at a speed of 2° per second. The instruction was “Press the button when the bar is aligned with a virtual plane dividing your corporeal space into two equal parts” (Figure 2 and Appendix A).

During the control condition, the same sequence of bar movements was presented, but participants were instructed to press the button when the bar changed direction.

Each fMRI sequence lasted 3 min and followed a block design: 30 s blocks of task and 30 s block of control conditions. In each block, the bar passed through the center six times at a constant speed of two degrees per second [20]. The process is similar for both tasks. The tasks were implemented using E-Prime 2.0 Professional software (Psychology Software Tools, Pittsburgh, PA, USA) and presented using goggles (NordicNeuroLab, Bergen, Norway) attached to the head coil to immerse participants in the task without any spatial clues. The instruction was presented via the goggles just before the task, in different colors for task and rest at the beginning of each block. To perform the tasks, participants reported their perception by pressing a button with their right index finger. Before the experiment, they practiced outside the scanner until they were able to perform the tasks. They all confirmed they understood the instructions. The difference between target identification (respectively, straight ahead or longitudinal axis) and button press time were computed both for SSA and SLB tasks.

### 2.3. MRI Acquisition 

MRI acquisition was performed on a 3 tesla scanner (Magnetom Verio; Siemens Medical Solutions, Erlangen, Germany) with a 12-channel head coil. The alignment of head, shoulders, trunk, and limbs was checked by the experimenter before MRI acquisition. The head and the trunk were maintained with foam pads.

The following MR sequences were acquired: -sagittal morphological isotropic 3D T1 MPRAGE sequence (repetition time (TR)/inversion time (TI)/echo time (TE): 1900/900/2.26 ms, field of view (FOV) 256 × 256 mm^2^, 160 slices, 1 × 1 × 1 mm^3^ voxel size);-field map with two echo times for distortion correction; and-two blood oxygen level dependent (BOLD) fMRI sequences using a single-shot T2*-weighted EPI sequence (TR/TE: 3000/36 ms, 210 × 210 mm^2^, FOV, 2 × 2 × 4 mm^3^, voxel size, 24 slices). Interleaved slices were acquired parallel to the anterior commissure posterior commissure line with no gap.

In total, the experiment was completed in 15 min.

Images were preprocessed and analyzed using Statistical Parametric Mapping (SPM12; Wellcome Department of Imaging Neuroscience, University College London, London, UK [22]; revision 7487) implemented in MATLAB. Morphological images were segmented, corrected for bias, and normalized to Montreal Neurological Institute atlas space (MNI). Slice timing, spatial realignment, distortion correction, coregistration to the morphological images, and normalization were applied, followed by 6 mm full width at half maximum (FWHM) isotropic Gaussian kernel smoothing. The canonical hemodynamic response function and its temporal derivative were used for model estimation, and three contrasts were evaluated: “SSA > control condition”, “SLB > control condition”, and “SLB > SSA”.

### 2.4. Statistical Modeling and Inference Analysis

Given that a large number of regions of interest are reported in the literature, we ran voxelwise analyses across the whole brain. A cluster extent of 25 voxels was determined by running a Monte Carlo simulation [23,24,25] using cluster threshold betas based on an individual voxel threshold of *p* = 0.001 and a corrected threshold of *p* = 0.05. The two tasks were compared using a paired t test with a cluster extent of 25 voxels.

To allow us to compare our results with those of previous studies [4,7,20,26,27,28,29], we converted the literature data into MNI coordinates using the MNI to Talairach application [30,31]. We visualized our results using SPM12 and the xjView toolbox (available online: https://www.alivelearn.net/xjview (accessed on 27 May 2021) and MRIcroGL (available online: https://www.mccauslandcenter.sc.edu/mricrogl/ (accessed on 27 May 2021). The interpretation was performed in MNI space, using the automated anatomical labeling atlas (AAL3) [32], Jülich brain cytoarchitectonic atlas [33,34,35,36], and brainnetome atlas [37] for cyto-myelo labeling. The latter, based on anatomical connectivity parcellation and containing 246 regions of the bilateral hemispheres (210 cortical and 36 subcortical subregions), was the most useful one for identifying activated regions reported in the results.

## 3. Results

Subjects responded within the standard to both tasks showing the reliability of the tasks. The SSA mean deviation was −0.4°, SD 1.3; the SLB mean deviation was −1°, SD 0.3. The mean response times of both tasks were not significantly different suggesting a similar complexity (*p* = 1.17).

### 3.1. Subjective Straight-Ahead Task (SSA)

The “SSA > control condition” contrast highlighted a parieto-occipital network shown in in Figure 3 and Table 1. We observed right-hemispheric dominance (74.3% of activated voxels).

The activated voxels in the eight significant clusters were distributed as follows: -the right parietal lobe (superior parietal lobule: A7c, A5l, A7pc; inferior parietal lobule: A39c, A40rd, A40rv; precuneus: dmPOS; postcentral gyrus: A2) representing 45.5% of activated areas;-the right lateral occipital cortex (msOccG, lsOccG, mOccG), representing 21.5% of activated areas;-the left lateral occipital cortex (V5/MT+, msOccG, lsOccG, mOccG), representing 16% of activated areas;-the left parietal lobe (superior parietal lobule: A5l, A7pc; and inferior parietal lobule: A39c), representing 9.5% of activated areas; and-the right premotor cortex (superior frontal gyrus: A6cdl), representing 7.5% of activated areas.

### 3.2. Subjective Longitudinal Body Plane Task (SLB)

The “SLB > control condition” contrast highlighted a fronto-parieto-occipital network extending to the temporal lobe and insula, with right-hemispheric predominance (75.5% of clusters located in right hemisphere). Activated clusters are shown in Figure 4 and Table 2.

Activated voxels were distributed as follows:-the right parietal lobe (superior parietal lobule: A7r, A7c, A5l, A7pc, A7ip; inferior parietal lobule: A39c, A40rd, A40rv; precuneus: dmPOS; postcentral gyrus: A1/2/3ulhf, A2, A1/2/3tru), representing 27% of activated areas;-the right lateral occipital cortex (V5/M+, msOccG, lsOccG, mOccG, iOccG), representing 22.5% of activated areas;-the right frontal lobe (predominant in the precentral gyrus: A4hf, A6cdl, A4ul, A4tl, A6cvl; superior frontal gyrus: A8m, A6dl, A6m; inferior frontal gyrus: A44d, A44op; paracentral gyrus: A4ll), representing 21% of activated areas;-the left lateral occipital cortex (V5/MT+, mOccG, iOccG), representing 8% of activated areas;-the left frontal lobe (supplementary motor area: A6m; precentral gyrus: A4hf, A6cdl, A4ul, A4t; and paracentral lobule: A4ll), representing 7.5% of activated areas; and-less than 5% in each of the following areas: left parietal lobe (superior parietal lobule: A5l, A7pc; inferior parietal lobule: A39c, A39rv; postcentral gyrus: A1/2/3ulhf, A2), right insular lobe (vIa, dIa, dIg, dId), left temporal lobe (middle temporal gyrus: A37dl; inferior temporal gyrus: A37vl; fusiform gyrus: A37lv), left cingulate gyrus, right temporal lobe (middle temporal gyrus: A37dl; inferior temporal gyrus: A37vl; fusiform gyrus: A37lv), and right cingulate gyrus.

### 3.3. Subjective Longitudinal Body Plane Task versus Subjective Straight-Ahead Task

Subjects responded within the standard to both tasks showing the reliability of the tasks. The SSA mean deviation was −0.4°, SD 1.3; the SLB mean deviation was −1°, SD 0.3. The mean response times of both tasks were not significantly different suggesting a similar complexity (*p* = 1.17).

Both tasks elicited a parieto-occipital network that encompassed the superior and inferior parietal lobules, precuneus, and lateral occipital cortex. The “SLB > SSA” comparison revealed a larger activation area (2055 vs. 633 voxels) and higher intensity for the SLB task than for the SSA task. Moreover, the “SLB > SSA” contrast revealed activation in the left angular and precentral gyri (Figure 5, Table 3).

## 4. Discussion

Using fMRI, the present study investigated the neural bases of processes elicited by two different egocentric tasks in healthy participants: SSA and SLB. This was the first time that the SLB task had been implemented in fMRI, reproducing the clinical assessment setting in a supine position without visual clues. We demonstrated its feasibility and complementarity with the SSA task [20]. These tasks can both be administered to patients with spatial cognitive disorders following stroke. Our results will enhance understanding of the egocentric reference frame concept both in healthy participants and in patients with a brain lesion.

The SSA and SLB both activated a parieto-occipital network with right-hemispheric predominance. This is in agreement with previous human and animal studies reporting activation of a fronto-parieto-occipital network by spatial cognition tasks [3,4,7,20,26,27,28,29,38,39,40]. Studies in nonhuman primates have highlighted the involvement of the posterior parietal cortex, parieto-insular vestibular cortex, and parieto-occipital sulcus as multimodal integration areas [38,39,40,41,42,43,44,45,46]. In 2010, Galati et al. published a review of studies investigating egocentric and allocentric tasks in healthy participants. This review, similar to more recent articles in humans [4,7,18,20,26,27,28,29], described the crucial role of the bilateral fronto-parietal network for spatial tasks, with involvement of the posterior parietal cortex extending to the superior frontal region [7,20,27,29]. In particular, egocentric activity was related to the posterior parietal cortex including the precuneus [27,29], superior parietal lobule [7,20,27,28], and intraparietal sulcus [20]. In the right hemisphere, activation maps often extend to the supramarginalis gyrus and angular gyrus [7,20,27]. These areas involved in the integration and selection of multimodal sensory inputs may, therefore, support the elaboration of mental representations of the body in space [8,20]. Activated areas reported in the superior and inferior parietal lobules confirm the involvement of somesthesis and motor imagery in the egocentric reference frame.

We found strong activity in the right lateral occipital cortex implicated in visual perception, corroborating previous experimental findings [43]. As we used a moving bar instead of a static image, the motion may have increased occipital activity.

Along these lines, we found less extensive frontal activation in the superior and inferior frontal gyri than in previous studies [27,29]. However, these studies compared an egocentric task with an allocentric task in the same run. As the background required to perform the allocentric task is similar in both the SSA and SLB conditions, participants may engage attention and inhibition functions in order to be able to shift between the two tasks, thereby, increasing cognitive load and thus prefrontal activity. This may explain the lack of prefrontal activity in our study, in which we used a separate background for each task.

By analyzing and comparing two tasks involving the egocentric reference frame, our study represents a milestone in the comprehension of egocentric frame mechanisms. We found significant differences in fMRI activity between the corporeal and extracorporeal egocentric tasks. Interestingly, the SLB task activated additional areas in the left hemisphere, including the angular (A39rd, A39rv, A40c) and precentral gyrus (A6cdl, A4ul). The angular gyrus is a high-level cognitive area involved in attentional and memory processes [47,48], whereas the premotor cortex is engaged in movement selection and preparation [49]. The left lateralization of premotor cortex activity during the SLB task can be viewed from the perspective of motor imagery tasks that typically activate the left premotor cortex [50,51,52]. This sheds light on the complexity of corporeal task processes. Moreover, the SLB task appears to require mental spatial coordinate transformation, as participants have to mentally align the external cues (the light bar) with their body’s midsagittal plane [53]. This view is supported by studies showing the implication of left parietal temporal occipital junction lesions in body schema disturbances, causing difficulties in spatial orientation [47]. Moreover, our results are consistent with the hypothesis of a functional interhemispheric connection involved in egocentric processing, particularly during corporeal tasks [15].

The present study had several limitations. First, the sample size is small and increasing sample sizes are desired. However, previous studies have included even smaller samples (fewer than 10 participants). Moreover, the age range is wide, but we do not expect variability in healthy controls as regards this kind of task [54,55,56]. Indeed, age has been reported to affect allocentric tasks while preserving egocentric strategies. Furthermore, we did not observe any age effect on task performance. Second, we focused our investigation on the egocentric reference frame in particular. Given the lack of study on this subject, we aimed to improve knowledge of the elaboration of the egocentric reference frame. Thus, we implemented two egocentric tasks oriented toward corporal or extracorporeal spaces with a control condition when the bar changed direction. In order to verify how specific the MRI findings are in terms of brain processing of egocentricity, it would have been interesting to add an allocentric task and compare our results to previous works. Third, the 3 min runtime may have been too short to obtain reliable results. However, we used an AB block design to compensate for the short duration of the tasks, which was a requirement in this clinical setting involving patients with stroke.

## 5. Conclusions

This study investigated two egocentric tasks oriented toward either extracorporeal or corporeal spaces in 17 healthy participants. There was a large overlap between the SSA and SLB tasks, with a right fronto-parieto-occipital pattern of activity, showing that they share a common network. However, the SLB task elicited additional activity in the left angular gyrus and premotor cortex, suggesting greater complexity for SLB processes with functional interhemispheric connectivity. These results shedding light on the clinical dissociations reported for spatial cognitive disorders need to be confirmed in studies involving patients with spatial cognition disorders.

## Figures and Tables

**Figure 1 brainsci-11-00963-f001:**
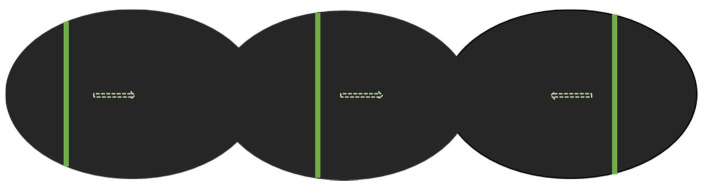
Schematic diagram of the subjective straight-ahead (SSA) fMRI task. Dotted arrows represent the direction of motion.

**Figure 2 brainsci-11-00963-f002:**
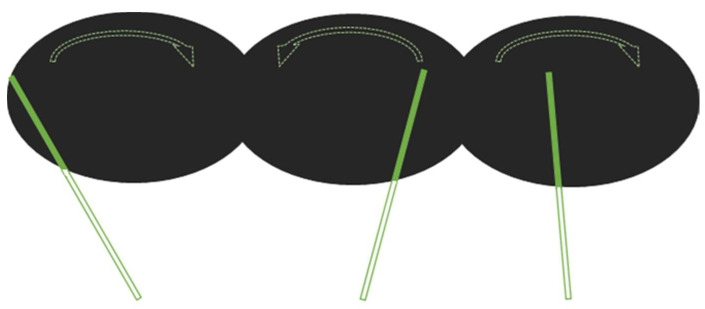
Schematic diagram of the subjective longitudinal body fMRI task. The dotted arrow represents the bar’s motion. The empty bars represent the virtual continuous plane centered on the navel.

**Figure 3 brainsci-11-00963-f003:**
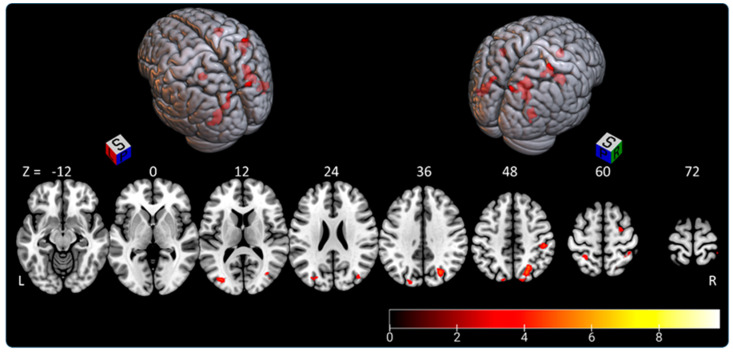
Rendered image of activated clusters highlighted by the “SSA > control condition” contrast. The color bar indicates *t*-values (BOLD); L—left, P—posterior, R—right, S—superior, Z coordinates in MNI atlas.

**Figure 4 brainsci-11-00963-f004:**
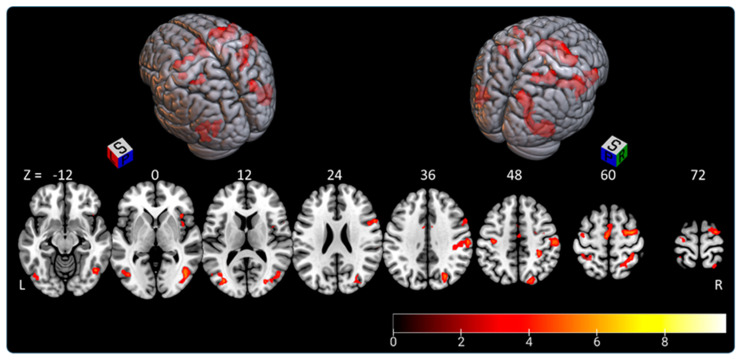
Rendered image of activated clusters highlighted by “SLB > control condition” contrast. The color bar indicates *t*-values (BOLD); L—left, P—posterior, R—right, S—superior, Z—coordinates in MNI atlas.

**Figure 5 brainsci-11-00963-f005:**
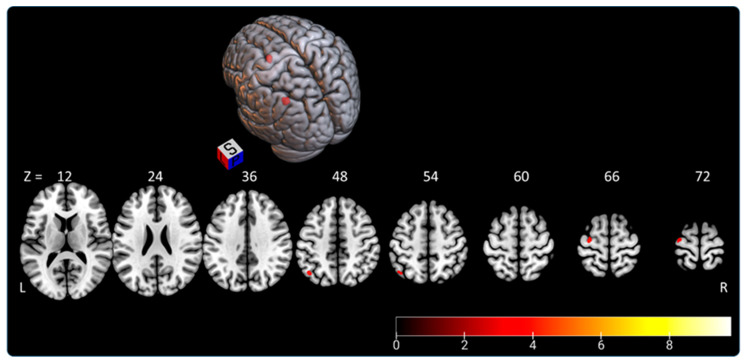
Rendered image of activated clusters highlighted by “SLB > SSA” contrast. The color bar indicates *t*-values (BOLD); L—left, P—posterior, R—right, S—superior, Z coordinates in MNI atlas.

**Table 1 brainsci-11-00963-t001:** Activated areas highlighted by the “SSA > control condition” contrast. The number of voxels is given for each cluster, together with the side (i.e., left or right hemisphere), the cluster composition (i.e., percentage of voxels in each area assigned using AAL3 and brainnetome atlas), MNI coordinates and t value of the peak. Only activated areas covering more than 10% of the cluster are shown.

Cluster Voxels/Side	AAL3 % in the Area	Peak MNI Coordinates (x y z)/T Value	Brainnetome % in the Area
226/right	42.5 occipital sup 33.6 parietal sup	24 −62 46/5.630	31.9 lsOccg 28.8 A7c
92/left	96.7 occipital mid	−30 −84 14/5.023	44.6 mOccG
			35.9 A39c
81/right	61.7 postcentral	46 −28 42/5.440	69.1 A40rd
	29.6 supramarginal		19.8 A2
66/right	87.9 occipital mind	42 −80 18/5.471	54.5 A39c 31.8 mOccG
58/right	55.2 postcentral 25.9 parietal inf 19.0 parietal sup	38 −44 66/5.216	12.1 A7pc 60.3 A5l 19.0 A7ip
40/right	72.5 frontal sup 17.5 precentral	26 −4 58/5.087	85.0 A6cdl 10.0 A6dl
39/left	76.9 occipital sup 23.1 parietal sup	−18 −86 38/4.506	10.3 msOccG 79.5 lsOccG
31/left	51.6 parietal sup	−26 −46 58/6.259	19.4 A7pc
	25.8 parietal inf 22.6 postcentral		38.7 A5l

**Table 2 brainsci-11-00963-t002:** Activated areas highlighted by “SLB > control condition” contrast. The number of voxels is given for each cluster, together with the side (i.e., left or right hemisphere), the cluster composition (i.e., the percentage of voxels in each area assigned using AAL3 and brainnetome atlas), MNI coordinates and t value of the peak. Only activated areas covering more than 10% of the cluster are shown.

Cluster Voxels/Side	AAL3 % in the Area	Peak MNI Coordinates (x y z)/T Value	Brainnetome % in the Area
565/right	25.3 occipital mid 25.1 temporal mid 18.9 occipital sup 13.3 temporal inf	44 −66 −6/10.448	16.6 mOccG 9.7 msOccG 8.5 lsOccG 38.1 V5/MT+
512/right	59.4 postcentral 16.8 supramarginal 15.0 parietal sup	50 −22 42/6.4996	18.8 A2 24.2 A40rd 15.4 A5l 11.1 A40rv
240/right	64.6 frontal sup	36 −6 58/6.921	56.3 A6cdl
	15.0 precentral 11.3 frontal mid		22.5 A6dl
218/left	59.2 occipital mid 20.2 temporal mid 18.4 occipital inf	−34 −78 10/6.0758	19.7 mOccG 45.4 V5/MT+ 11.0 A39c
152/right-left	59.9 supp motor area L 31.6 sup motor area R	−4 −12 58/6.0816	51.3 lA6m 26.3 rA6m
107/right	60.8 frontal inf oper 39.3 precentral	58 12 30/6.2155	92.5 A6cvl
78/right	87.2 insula	44 20 −6/5.5758	34.6 dla 44.9 dld 11.5 A44op
55/left	98.2 precentral	−40 −10 66/5.086	45.5 A6cdl 29.1 A4hf 18.2 A4ul
52/left	73.1 postcentral 26.9 parietal sup	−40 −44 62/4.699	36.5 A7pc 38.5 A5l 19.2 A2
45/right-left	53.3 cingulate mid L 31.1 cingulate mid R 11.1 sup motor area L	−2 6 42/6.001	57.8 lA24cd 35.6 rA24cd
31/left	51.6 postcentral	−38 −18 46/5.1726	38.7 A1/2/3ulhf
	48.4 precentral		41.9 A4hf

**Table 3 brainsci-11-00963-t003:** Activated areas highlighted by “SLB > SSA” contrast. The number of voxels is given for each cluster, together with the side (i.e., left or right hemisphere), the cluster composition (i.e., the percentage of voxels in each area assigned using AAL3 and brainnetome atlas), MNI coordinates and t value of the peak. Only activated areas covering more than 10% of the cluster are shown.

Cluster Voxels/Side	AAL3 % in the Area	PeakMNI Coordinates (x y z)/T Value	Brainnetome % in the Area
28/left	71.0 angular 7.1 parietal inf	−44 −66 50/4.9301	57.1 A39rd 7.1 A39rv
25/left	92.0 precentral	−30 −14 70/6.430	52.0 A6cdl
			40.0 A4ul

## Data Availability

The datasets used and analyzed during the current study are available from the corresponding author on request.

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
