# Peer review of "The Neural Bases of Egocentric Spatial Representation for Extracorporeal and Corporeal Tasks: An fMRI Study"

_brainsci, 2021, doi:10.3390/brainsci11080963_

Round 1

Reviewer 1 Report

Summary. The aim of this fMRI study was to verify the existence of a dissociation at neural level between the extracorporeal subjective straight-ahead line (SSA) and the subjective longitudinal body plane line (SLB) in humans. The results showed that both of these processes involved a right occipito-temporal network, while the angular gyrus and the premotor cortices were more involved during the task done to measure the SLB.

Comments. The work's strengths include novelty. However, there are some very critical points that if not clarified properly could lead me to reject the work later. The points are listed below:
1. I have difficulty in grasping the difference between SSA and SLB with respect to the much better known distinction in the literature between extrapersonal and peripersonal space. I invite the authors to clarify this distinction;
2. The distinction between the two tasks (SSA and SLB) in the Introduction is not clear, it speaks for both of them of dividing the space on the basis of the egocentric reference system, always using the same subjective line/midline. How can a participant be expected to make a difference between a "subjective straight-ahead" and a "subjective longitudinal doby plane"? This difference needs to be clarified very well both in theoretical and operational terms. In fact the two tasks could be carried out using the same strategy, it is only the instructions that change. This crucial point could undermine the goodness of the work. The fact that the authors found additional areas that are activated during the SLB task certainly does not justify a priori the difference between the two tasks;
3. It is not clear from where the authors derive their hypotheses. Why do they expect those activations for the two tasks (i.e. SSA and SLB)? Help might come by referring to Ruotolo and colleagues' 2019 fMRI work, in which subjects were asked to take their body-midline as a reference and based on this make metric (more action-related) and categorical (more recognition-related) judgments. The brain areas involved seem to be similar to those expected by the authors and would therefore receive further support in their study
4. In the methods section much more detail needs to be included to allow replicability of the study, as well as greater understanding by the reader. For example, how many trials? How many blocks? How were they administered? How long was the experiment in total?
5. Another rather worrying point. Unfortunately, the tasks lack a measure of accuracy and/or response time that could in some way measure the performance of the participants. It is certainly not enough to say that "We monitored participants on a screen in real time, to ensure that they performed the task (lines 113-114)" to ensure that participants actually performed the two tasks according to the instructions. Without these measures we could never know what the differences in activation between the two tasks depended on. Was one task more difficult than the other? This is a very serious point.
6. In the conclusions it is said that the study is exploratory, but this is not the case. There is sufficient evidence in the literature to formulate precise hypotheses.

Minor points:
Is this sentence in the Abstract necessary?: "This work preceded a study involving stroke patients", Isn't the study involving patients?

Reviewer 2 Report

General comment.

As this is claimed to be the first study addressing the egocentric SSA and SLB tasks in fMRI, I guess that a control condition with an allocentric task (maybe using the same visual stimuli used here) is necessary, in order to verify how specific are the MRI findings in terms of brain processing of egocentricity.

A second point is that, if the aim of the study is to build up a data base of normal subjects to be matched with stroke patients in a forthcoming study (see discussion), the sample is really small, also considering the  wide age range (23-70 years) of the population.

Other points are:

Handedness of subjects should be provided and quantified.

The description of the SSA and SLB tasks are not clear to me: in the SSA, it is said that subjects were asked to imagine a plane in front of them, then a black screen with a moving green vertical line is shown (fig. 1). In SSB, the same screen is displayed, and another green line (different from the previous one) moved as a reversed pendulum. So, it seems that they are mixing imagery and visual tasks, but with different visual stimuli. Finally, as active and control tasks were intermingled, how subjects identified when the task was active or control?

Round 2

Reviewer 1 Report

Unfortunately, the authors did not respond satisfactorily to the criticisms made. There remains a crucial critical point for all fMRI work and that is to be able to attribute the increase in brain activation to the task and not to its difficulty. 

Reviewer 2 Report

I guess that the basic weaknesses of the study I mentioned in the previous round of review (as general comments) have not been answered.
